# *Bovine Polyomavirus-1* (*Epsilonpolyomavirus bovis*): An Emerging Fetal Pathogen of Cattle That Causes Renal Lesions Resembling Polyomavirus-Associated Nephropathy of Humans

**DOI:** 10.3390/v14092042

**Published:** 2022-09-14

**Authors:** Federico Giannitti, Caroline da Silva Silveira, Hannah Bullock, Marina Berón, Sofía Fernández-Ciganda, María José Benítez-Galeano, Nélida Rodríguez-Osorio, Luciana Silva-Flannery, Yisell Perdomo, Andrés Cabrera, Rodrigo Puentes, Rodney Colina, Jana M. Ritter, Matías Castells

**Affiliations:** 1Plataforma de Investigación en Salud Animal, Instituto Nacional de Investigación Agropecuaria (INIA), Estación Experimental La Estanzuela, Colonia 70006, Uruguay; 2Synergy America Inc., Atlanta, GA 30329, USA; 3Unidad de Genómica y Bioinformática, Departamento de Ciencias Biológicas, Centro Universitario Regional (CENUR) Litoral Norte, Universidad de la República, Salto 50000, Uruguay; 4Infectious Diseases Pathology Branch, Centers for Disease Control and Prevention (CDC), Atlanta, GA 30329, USA; 5Facultad de Veterinaria, Universidad de la República, Montevideo 13000, Uruguay; 6Laboratorio de Interacciones Hospedero-Patógeno, Institut Pasteur de Montevideo, Montevideo 11400, Uruguay; 7Laboratorio de Virología Molecular, Departamento de Ciencias Biológicas, Centro Universitario Regional (CENUR) Litoral Norte, Universidad de la República, Salto 50000, Uruguay

**Keywords:** abortion, cattle, pathology, reproductive diseases, viral diseases, emerging diseases, polyomavirus, nephropathy, *Epsilonpolyomavirus bovis*, next generation sequencing

## Abstract

*Bovine polyomavirus-1* (BoPyV-1, *Epsilonpolyomavirus bovis*) is widespread in cattle and has been detected in commercialized beef at supermarkets in the USA and Germany. BoPyV-1 has been questioned as a probable zoonotic agent with documented increase in seropositivity in people exposed to cattle. However, to date, BoPyV-1 has not been causally associated with pathology or disease in any animal species, including humans. Here we describe and illustrate pathological findings in an aborted bovine fetus naturally infected with BoPyV-1, providing evidence of its pathogenicity and probable abortigenic potential. Our results indicate that: (i) BoPyV-1 can cause severe kidney lesions in cattle, including tubulointerstitial nephritis with cytopathic changes and necrosis in tubular epithelial cells, tubular and interstitial inflammation, and interstitial fibroplasia; (ii) lesions are at least partly attributable to active viral replication in renal tubular epithelial cells, which have abundant intranuclear viral inclusions; (iii) BoPyV-1 large T (LT) antigen, resulting from early viral gene expression, can be detected in infected renal tubular epithelial cells using a monoclonal antibody raised against Simian Virus-40 polyomavirus LT antigen; and (iv) there is productive BoPyV-1 replication and virion assembly in the nuclei of renal tubular epithelial cells, as demonstrated by the ultrastructural observation of abundant arrays of viral particles with typical polyomavirus morphology. Altogether, these lesions resemble the “cytopathic-inflammatory pathology pattern” proposed in the pathogenesis of *Human polyomavirus-1*-associated nephropathy in immunocompromised people and kidney allograft recipients. Additionally, we sequenced the complete genome of the BoPyV-1 infecting the fetus, which represents the first whole genome of a BoPyV-1 from the Southern Hemisphere. Lastly, the BoPyV-1 strain infecting this fetus was isolated, causing a cytopathic effect in Madin–Darby bovine kidney cells. We conclude that BoPyV-1 is pathogenic to the bovine fetus under natural circumstances. Further insights into the epidemiology, biology, clinical relevance, and zoonotic potential of BoPyV-1 are needed.

## 1. Introduction

Polyomaviruses are a diverse group of non-enveloped viruses with a small, circular, double-stranded DNA genome found in a wide variety of mammalian (including humans), avian, fish, amphibian, reptile, and invertebrate species [1]. As of September 2022, according to the International Committee on Taxonomy of Viruses (ICTV, https://talk.ictvonline.org/, accessed on 13 September 2022), the *Polyomaviridae* family contained 8 genera named *Alpha*-, *Beta*-, *Delta*-, *Epsilon*-, *Eta*-, *Gamma*-, *Theta*- and *Zeta-polyomavirus*, although many strains and species are awaiting genus assignation. While some polyomaviruses can cause acute disease and even death due to productive (lytic) replication in their hosts, most of the polyomaviruses infecting mammals establish persistent subclinical infections in healthy individuals, resulting in clinical disease only after reactivation of the infection in immunosuppressed hosts [2].

Most information on the biology and medical relevance of polyomaviruses has been generated in studies of primate and murine polyomaviruses; limited information is available from other mammalian polyomaviruses in their natural hosts. Prototype diseases caused by human polyomaviruses include “polyomavirus-associated nephropathy” (PyVAN) caused by *Human polyomavirus-1* (BK polyomavirus –BKPyV–, *Betapolyomavirus hominis*) and to a lesser extent *Human polyomavirus-2* (JC polyomavirus –JCPyV–, *Betapolyomavirus secuhominis*) in kidney transplant recipients; and “progressive multifocal leukoencephalopathy” (PML) resulting from replication of JCPyV in oligodendrocytes in patients with acquired immunodeficiency syndrome or other immunomodulatory conditions [1]. Polyomaviruses have also been associated with cancer in humans and animals. For example, *Human polyomavirus-5* (Merkel cell polyomavirus –MCPyV–, *Alphapolyomavirus quintihominis*) causes Merkel cell (neuroendocrine) carcinoma of the skin [3], while *Procyon lotor polyomavirus-1* (syn. *Raccoon polyomavirus-1* –RacPyV-1–, *Alphapolyomavirus procyonis*) has been associated with tumors of the olfactory tract and brain in raccoons [4,5,6]. A recent investigation found that human females affected by spontaneous abortion have significantly lower levels of IgG serum antibodies against the oncogenic MCPyV than those undergoing voluntary pregnancy interruption [7]. While there is no clear association with abortion causality, the authors hypothesized that this immunological decrease might prompt an increase in MCPyV multiplication events in females experiencing spontaneous abortive events [7].

The polyomaviruses currently known as bovine polyomavirus (BoPyV) were initially discovered in uninoculated cultures of kidney cell lines of stump-tailed macaques, rhesus monkeys, and cynomolgus macaques, and were referred to as Stump-Tailed Macaque Virus (STMV) [8], cynomolgus kidney strain (CK-strain) of STMV [9], and Fetal Rhesus Kidney Virus (FRKV) [10]. Because of their growth in uninoculated cell cultures, it was initially thought that they were endogenous viruses of non-human primates. However, it has since been determined that they were polyomaviruses of bovine origin contaminating the bovine fetal serum used to supplement cell growth media [11]. Such contamination is frequent in commercial batches of bovine fetal/calf serum [12,13,14,15,16]. The first polyomavirus isolated in a bovine kidney cell line was obtained from a healthy newborn calf [17] and was designated Wokalup Research Station Virus (WRSV). It was suggested shortly after their discovery that STMV, FRKV, and WRSV were all isolates of an identical BoPyV [18], which was later supported by viral whole-genome sequencing of the isolate obtained by Wognum et al. [19].

Species of polyomaviruses known to infect live cattle to date include *Bos taurus polyomavirus-1* (syn. *Bovine polyomavirus-1* –BoPyV-1–, *Epsilonpolyomavirus bovis*), and *Bos taurus polyomavirus-2* (syn. *Bovine polyomavirus-2* –BoPyV-2–, awaiting genus assignation). A third species named *Bovine polyomavirus-3* (BoPyV-3, awaiting genus assignation) was originally detected in ground beef samples collected at a supermarket in the USA [20]. The complete genome sequence for BoPyV-3 is deposited in GenBank (accession KM496326), but the species is not currently listed by the ICTV. To the best of our knowledge, BoPyV-1 and BoPyV-3 have never been causally associated with pathology or disease, while BoPyV-2 has been recently proposed as a probable cause of nonsuppurative encephalitis in cattle [21].

Based on a serologic study in humans, published by authors from the United Kingdom, BoPyV has been questioned as a zoonotic agent with documented increase in seropositivity in people occupationally exposed to cattle, including veterinarians, farmers, abattoir workers, veterinary technical staff, and veterinary students [18]. Although these results should be interpreted with caution due to possible cross-reaction of human antibodies to other polyomaviruses [22,23], the risk of zoonotic transmission of BoPyV should not be neglected and deserves further research.

Abortion is a major health problem in cattle, resulting in huge economic losses to the livestock industry worldwide. It can be caused by many infectious and non-infectious diseases. Infectious etiologies are amongst the most frequently detected causes of abortion in ruminant fetuses subjected to routine laboratory diagnostic investigation, and include a variety of protozoal, bacterial, viral, and fungal pathogens, many of which are zoonotic [24,25]. Because many bovine pathogens can be transmitted transplacentally from the dam to the fetus without necessarily resulting in abortion, detecting an infectious agent in the fetus does not warrant abortion causality. However, pathogen detection coupled with the identification of pathogens within lesions observed on histopathologic examination of the aborted fetus and/or placenta is a powerful indicator of causality [26]. Given the large spectrum of possible abortigenic pathogens and the relatively few veterinary diagnostic laboratories that conduct pathologic examinations with broad pathogen detection in aborted ruminants, especially in low- and middle-income countries, it is generally accepted that known causes of abortion are underreported and that many abortifacients remain to be discovered.

The aim of this study was to describe and illustrate pathological findings in an aborted bovine fetus naturally infected with BoPyV-1, providing strong evidence to consider it fetopathogenic and a probable cause of abortion in cattle. We also sequenced the complete genome of the involved BoPyV-1, which, to the best of our knowledge, represents the first sequence of a BoPyV-1 from the Southern Hemisphere.

## 2. Materials and Methods

### 2.1. Histopathology and Immunohistochemistry

Samples of fetal brain, kidney, liver, heart, spleen, lung, trachea, esophagus, tongue, skeletal muscle, eyelid/conjunctiva, lymph node, adrenal gland, abomasum, forestomachs, thymus, testicle, and small and large intestines were immersion-fixed in 10% neutral buffered formalin, processed, embedded in paraffin, microtome-sectioned, and stained with hematoxylin and eosin for histopathology at the veterinary laboratory of the “Instituto Nacional de Investigación Agropecuaria” (INIA), Colonia, Uruguay. Sections were examined by a veterinary pathologist under an optical microscope (Axio Scope.A1, Carl Zeiss, Göttingen, Germany) coupled with a color digital camera (Axiocam 512, Carl Zeiss, Germany) commanded by the ZEN software (Carl Zeiss, Germany).

Sections of liver and kidney were processed by immunohistochemical assays for the detection of viruses known to produce intranuclear inclusion bodies in epithelial cells, including polyomavirus, herpesvirus, and adenovirus. For polyomavirus, tissues were subjected to heat-induced epitope retrieval in citrate buffer, and a mouse IgG2a monoclonal antibody raised against Simian Virus-40 (SV-40, *Macaca mulatta polyomavirus-1*, *Betapolyomavirus macacae*) LT antigen (CalBiochem^®^, Clone PAb416, Millipore, Sigma-Aldrich, Saint Louis, MO, USA) was used as primary antibody at a 1:200 dilution. Colorimetric detection of linked antibodies was performed using the Mach 4 AP Polymer kit (Biocare Medical, Concord, CA, USA) followed by visualization with Permanent Red Chromogen (Cell Marque^TM^, Millipore-Sigma-Aldrich, Rockling, CA, USA). Slides were counterstained with Mayer’s hematoxylin (Poly Scientific, Bay Shore, NY, USA) and coverslipped with aqueous mounting medium (Polysciences Inc., Warrington, PA, USA).

Two immunohistochemical assays were performed for the detection of herpesviruses. For one, antigen retrieval was accomplished by treatment with proteinase K, and a rabbit polyclonal antibody raised against human herpesvirus-1 (CDC Biological Products) that cross-reacts with human herpesvirus-2 was applied at a dilution of 1:3000. The same revealing system as described for the SV-40 immunohistochemistry was used. The other assay was performed as previously described [27] using a primary antibody against bovine herpesvirus-1.

For adenovirus immunohistochemistry, antigen retrieval was accomplished by treatment with 0.4% pepsin and a primary antibody raised against deer adenovirus that cross-reacts with bovine adenovirus was applied. This assay was performed following a previously described procedure [28] with minor modifications (the Dako Envision + system, rather than the Biocare Farma, was used as the detection system).

For each of the immunohistochemical assays, appropriate positive and negative controls were used in parallel for quality assurance purposes and identification of nonspecific immunoreactions. The immunohistochemical assays for polyomavirus and human herpesvirus-1 were conducted at the Centers for Diseases Control and Prevention (CDC), while the assays for bovine herpesvirus-1 and adenovirus were conducted at the California Animal Health & Food Safety laboratory (University of California, Davis).

### 2.2. Transmission Electron Microscopy 

Transmission electron microscopy was conducted in formalin-fixed paraffin-embedded sections of kidney, using the on-slide embedding method [29] at the CDC. Briefly, 4 µm thick sections of tissue affixed to glass slides were deparaffinized in xylene, then rehydrated and fixed in 2.5% buffered glutaraldehyde. Samples were post fixed with 1% osmium tetroxide, *en bloc* stained with uranyl acetate, dehydrated, and embedded in Epon-Araldite resin. Epoxy resin-embedded glass slide sections were immersed in boiling hot water, removed from the slides with a razor blade, and areas of interest were glued onto EM blocks. Ultrathin sections were stained with uranyl acetate and lead citrate and examined on a Thermo Fisher/FEI Tecnai BioTwin electron microscope.

### 2.3. Molecular Virology and Ancillary Testing for Specific Pathogens

Nucleic acids were extracted from kidney, liver, and brain samples from the aborted bovine fetus, using the MagMAX^TM^ CORE Nucleic Acid Purification Kit (Applied Biosystems, Thermo Fisher Scientific, Life Technologies Corp., Austin, TX, USA) at INIA. The extracted nucleic acids were initially processed by polymerase chain reaction (PCR) for the detection of bovine herpesviruses-1, -4, and -5 [30,31] and reverse-transcriptase PCR for bovine viral diarrhea virus (*Pestivirus*) [32,33] at “Universidad de la República” (UdelaR), and PCR for *Neospora caninum* [34] at INIA. DNA extracted from kidney and liver was also tested by a real-time PCR assay targeting the *lipL32* gene of pathogenic *Leptospira* spp. [35] at INIA. 

For polyomavirus detection, real-time PCR targeting a 77 bp fragment of the BoPyV VP1 gene was performed as described elsewhere [36] at UdelaR. Briefly, 12.5 µL of SensiFAST Probe No-ROX Kit (Bioline^®^, London, UK), 5.0 µL of nuclease-free water, 1.0 µL of 10 µM forward primer (QB-F1-1), 1.0 µL of 10 µM reverse primer (QB-R1-1), 0.5 µL of 10 µM probe (QB-P1-2), and 5 µL of DNA were mixed in 0.2-mL PCR tubes.

A partial BoPyV VP1 gene sequence (527 bp) was amplified from the DNA by a conventional PCR at UdelaR. Briefly, 12.5 µL of MangoMix^TM^ (Bioline^®^, London, UK), 4.5 µL of nuclease-free water, 1.0 µL of 10 µM forward primer (VP1-F), 1.0 µL of 10 µM reverse primer (VP1-R), 1.0 µL of dimethyl sulfoxide, and 5 µL of cDNA were mixed in 0.2-mL PCR tubes. Primers and PCR conditions were previously described [15]. The PCR product was visualized in 2% agarose gel, purified using PureLink^TM^ Quick Gel Extraction kit and PCR Purification Combo Kit (Invitrogen, Life Technologies, Carlsbad, CA, USA), and sequenced at Macrogen Inc. (Seoul, Korea) with Sanger technology.

Lastly, a sample of serum obtained from the dam at the time of abortion was processed by real-time PCR targeting the BoPyV VP1 gene as described above. 

### 2.4. Whole-Genome Sequencing and Genome Characterization

For BoPyV whole-genome sequencing, conducted at UdelaR, viral genomic DNA was purified from the whole genomic DNA from kidney by extracting the 4–6 kb region of a 1% agarose gel. DNA purity, integrity, and concentration were assessed with a Qubit device (Thermo Fisher Scientific). The sequencing library was prepared with the ligation sequencing kit (SQK-LSK109) following the manufacturer’s instructions and directly sequenced on a FLO-MIN106 flow cell in a MinION device (Oxford Nanopore Technologies^®^, Oxford, UK) for 24 h. High-accuracy basecalling was performed with Guppy v3.6.0, and reads were trimmed and filtered with Nanofilt and Nanoplot [37]. Reads with quality over 10 were used in further analyses. A host filtering step was executed by mapping clean reads to the *Bos taurus* reference genome (GCF_002263795.1_ARS-UCD1.2_genomic.fna) using Minimap2 [38]. Unmapped reads were then mapped against the BoPyV-1 reference genome (NC_001442) and the consensus sequence was obtained using SAMtools [39]. The obtained complete genome sequence was deposited in GenBank. Genome annotation was performed using the BoPyV-1 reference genome (NC_001442).

### 2.5. Phylogenetic Analyses

To classify the BoPyV, a phylogenetic tree was performed using LT antigen amino acid sequences. Representative sequences of all the ICTV recognized genera, all the polyomavirus sequences of bovine origin available in GenBank, and other *Epsilonpolyomavirus* genus sequences of non-bovine origin were used. For phylogenetic analysis of the complete genome, all the complete BoPyV genomes were retrieved from GenBank. Finally, all the available BoPyV sequences including LT antigen, ST antigen, VP1, VP2, and VP3 were used to perform additional phylogenetic trees with nucleotide and amino acid sequences, to deepen the molecular characterization.

Multiple sequence alignments were performed with Clustal Omega provided at the EMBL-EBI [40]. The evolutionary model for the data and a maximum-likelihood phylogenetic tree were inferred with W-IQTREE [41]. Branch support was estimated with the approximate likelihood-ratio test (1000 replicates) [42].

### 2.6. Sequence Identity

The alignments for complete genomes and of each individual gene and protein were used to obtain the sequence identity matrices with BioEdit version 7.2.6 [43].

### 2.7. Virus Isolation

A sample of fetal kidney stored in a freezer at −20 °C for nearly 1 year was thawed and processed for virus isolation at UdelaR. Briefly, 20 mg of sample were inoculated on Madin–Darby bovine kidney (MDBK) cells cultured in a sterile 24-well cell culture plate (Costar^TM^, Corning Inc. Life Sciences, Tewksbury, MA, USA) with Dulbecco’s minimum essential medium supplemented with 1% penicillin/streptomycin solution and 5% commercial gamma-irradiated fetal bovine serum (Sigma-Aldrich, USA). Cultures were kept at 37 °C in an atmosphere with 5% CO_2_ and observed daily under inverted microscope to search for cytopathic effect. Five days post-inoculation of the kidney, 100 µL of the supernatant was inoculated onto a new sterile plate with the same cell line, that was similarly cultured and observed daily for cytopathic effect (first passage). A second passage was performed by inoculating 100 µL of the supernatant obtained at 4 days post-inoculation from the first passage into a new plate that was similarly cultured and observed daily for cytopathic effect. Each passage was performed in triplicates, observing a cytopathic effect in each of the replicates. At the same time, uninfected cells were seeded onto wells in each of the plates as control of the cell culture and reagents. Aliquots of the supernatant obtained from each one of the three culture plates at 5-, 4-, and 5-days post-inoculation were processed for DNA extraction and BoPyV-1 real-time PCR as described in the molecular virology section.

## 3. Results

### 3.1. Case History

In March 2021, a second-gestation Holstein cow from a *Brucella*-free dairy farm in Colonia, Uruguay, aborted a male fetus at 234 days (~7.7 months) of gestation. The fetus and a sample of serum of the dam were submitted to INIA’s veterinary laboratory for pathologic examination.

### 3.2. Gross Pathologic Examination

At autopsy, the fetus was in good state of postmortem preservation, had a fully developed hair coat, and a crown-to-rump length of 65 cm. There was mild clear subcutaneous edema in the ventral aspect of the cervical region, diffuse petechiae in the thymus, and paintbrush hemorrhages in the serosa/adventitia of the intra-abdominal segments of the umbilical arteries, suggesting that the fetus was alive until shortly before expulsion. A moderate amount of red-tinged serous fluid was present in the abdominal, thoracic, and pericardial cavities.

Scattered throughout both kidneys there were numerous discrete pinpoint red foci with a widespread distribution alternating with areas of pallor that were visible from the capsular surface (Figure 1A), which on cut section corresponded with enhanced cortical and medullary rays in the renal parenchyma. The liver had multifocally extensive areas of roughness and white pale discoloration visible from the diaphragmatic capsular surface. On cut section, the underlying hepatic parenchyma had increased consistency and enhanced reticular pattern characterized by numerous pinpoint pale grayish foci and interconnected linear streaks with poorly distinct borders of approximately 1 mm width alternating with reddish-orange areas of the hepatic parenchyma, resembling the so-called “nutmeg liver” (Figure 2A).

### 3.3. Histopathology and Immunohistochemistry

The most striking microscopic lesions were in the kidneys and included severe widespread tubulointerstitial nephritis affecting, predominantly, the renal cortex, but also the medulla (Figure 1B–D). Affected tubules were variably ectatic, frequently contained necrotic eosinophilic cellular debris sloughed into their lumens and were lined by attenuated epithelium. Multifocally, tubular epithelial cells showed either tumefaction with swollen and vesicular nuclei frequently containing one or several round or pleomorphic basophilic viral inclusion bodies, or were shrunken with angular cell borders, hypereosinophilic cytoplasm, and pyknotic nucleus or karyorrhectic debris. In affected areas, the interstitium was multifocally infiltrated by large numbers of lymphocytes, histiocytes, plasma cells, and rare neutrophils, or expanded by spindle cells with an elongate nucleus (fibroblasts) embedded in an eosinophilic fibrillar collagenous extracellular matrix, consistent with fibroplasia.

In the liver, the histoarchitecture was distorted, the hepatic cords were multifocally disorganized and separated by interconnecting bands of connective tissue (dissecting fibrosis), which also mildly expanded the portal tracts and surrounded some centrilobular veins. There was multifocal random lymphocytic and histiocytic hepatitis with rare neutrophils, multifocal individual hepatocellular death (necrosis or apoptosis), and multifocal mild to moderate portal hepatitis (Figure 2B–D).

Other less severe lesions included mild multifocal infrequent gliosis in the brain with rare perivascular and leptomeningeal lymphocytic infiltrates, and mild megakaryocyte hyperplasia (suggestive of extramedullary hematopoiesis) in the spleen. No significant lesions were observed in the other examined tissues (see Section 2.1 for a list of examined tissues).

Based on the abundance of intranuclear viral inclusion bodies in the kidney, and the presence of severe histologic lesions in the liver, these two tissues were selected to perform immunohistochemical assays. The assay using a mouse monoclonal antibody raised against SV-40 LT antigen revealed abundant strong granular immunoreactivity in the kidney that was largely restricted to the renal tubules in both the cortex and medulla and was more intense in the nuclei of the epithelial cells (Figure 1E,F). No immunoreactivity was observed in the renal glomeruli. No polyomavirus antigen was detected in a section of liver. The immunohistochemical assays for the detection of herpesviruses and adenoviruses were negative in the kidney and liver. 

### 3.4. Transmission Electron Microscopy

Transmission electron microscopy evaluation revealed abundant electron-dense viral particles morphologically consistent with polyomavirus in the nuclei of renal tubular epithelial cells (Figure 3). Viral particles measuring between 35 and 43 nm in diameter were arranged in icosahedral arrays, creating inclusions within the nuclei.

### 3.5. Molecular Virology and Ancillary Testing for Specific Pathogens

The polyomavirus VP1 fragment was successfully amplified from the fetal kidney, liver, and brain tissues and serum of the dam by real-time PCR with Ct values of 6.3, 14.8, 19.9, and 20.0, respectively. Furthermore, a VP1 gene partial sequence (527 bp), amplified by conventional PCR and sequenced, confirmed the presence of BoPyV-1 in the fetus. PCRs for bovine herpesvirus-1, -4, and -5, and for bovine viral diarrhea virus (*Pestivirus*) were all negative. PCR for *N. caninum* was positive in the brain and negative in the kidney and liver. Real-time PCR for pathogenic *Leptospira* spp. was negative in the kidney and liver.

### 3.6. Whole Genome Sequencing and Genome Characterization

The complete BoPyV-1 genome assembly obtained in this work consisted of 4697 bp and was assembled from 4712 Oxford Nanopore Technology long reads with an average length of 1248 bp and 1090X coverage (Appendix A). The complete genome sequence, named BoPyV-1/Faber/2021/Uy, was deposited in GenBank with accession number OM938033. The nucleotide composition was 28.7%, 21.2%, 29.9%, and 20.1% of A, C, T, and G, respectively. The genomic organization was similar to that of other BoPyV-1 reported genomes, with an early coding region containing ORFs encoding two proteins (LT antigen and small T –ST– antigen) and a late coding region containing ORFs that encode 3 structural proteins (VP1, VP2, and VP3) and a viral agnoprotein (Figure 4).

### 3.7. Phylogenetic Analyses

The current classification system of polyomaviruses is based on the LT antigen amino acid sequence analysis. Based on this, we classified BoPyV-1/Faber/2021/Uy within the *Epsilonpolyomavirus* genus, closely related to the other BoPyV-1, which is currently named *Epsilonpolyomavirus bovis* (Figure 5A). We also performed additional analyses to deepen the phylogenetic description of this fetopathogenic strain. Based on the phylogenetic analysis using complete genomes, BoPyV-1/Faber/2021/Uy grouped with other BoPyV-1 sequences, closely related to D13942 and KM496323 strains (Figure 5B). Similar results were observed when the analyses were performed using LT antigen, ST antigen, VP1, VP2, and VP3, with nucleotide and amino acid sequences; BoPyV-1/Faber/2021/Uy was also closely related to other *Epsilonpolyomavirus bovis* (Appendix A).

### 3.8. Sequence Identity

At the nucleotide level (Table 1), based on the complete genome, LT antigen, ST antigen, VP1, and VP3, BoPyV-1/Faber/2021/Uy showed the highest similarity (99.3%, 99.3%, 99.7%, 99.3%, and 99.2%, respectively) to D13942, ranging between 93.6% and 99.7% to other *Epsilonpolyomavirus bovis* (BoPyV-1). In the ST antigen and VP3 genes, BoPyV-1/Faber/2021/Uy also shared the same highest similarity (99.7% and 99.2%, respectively) with KM496323. Based on the VP2 gene, BoPyV-1/Faber/2021/Uy was most similar to KU200259 (99.3%), varying between 97.0% and 99.3% from other *Epsilonpolyomavirus bovis* (BoPyV-1). The nucleotide similarity ranged between 26.5% and 55.0% to BoPyV-2, and 26.7% and 48.3% to BoPyV-3.

At the amino acid level (Table 1), based on the LT antigen, ST antigen, VP1, and VP2 proteins, BoPyV-1/Faber/2021/Uy was most similar (99.6%, 99.1%, 99.7%, and 99.7%, respectively) to D13942, varying between 94.1% and 99.7% to other *Epsilonpolyomavirus bovis* (BoPyV-1). In the VP1 protein BoPyV-1/Faber/2021/Uy also shared the same highest similarity (99.7%) with KU170643. Based on the ST antigen and VP2 proteins, BoPyV-1/Faber/2021/Uy was equally similar (99.1% and 99.7%, respectively) to all the other *Epsilonpolyomavirus bovis* (BoPyV-1) except KU200259 and KX455485, respectively. Lastly, based on the VP3 protein, BoPyV-1/Faber/2021/Uy showed 100% similarity with the other *Epsilonpolyomavirus bovis* (BoPyV-1) except KX455485 (96.1%). The amino acid identity of BoPyV-1/Faber/2021/Uy ranged between 15.7% and 52.2% to members of the BoPyV-2, and 13.0% and 36.9% to BoPyV-3.

### 3.9. Virus Isolation

Cytopathic effect characterized by cytoplasmic vacuolation followed by lysis of MDBK cells was observed 5 days after inoculation of the kidney tissue, and 4 and 5 days after inoculation of the supernatants on the two (first and second) serial passages (Figure 6A). The real-time PCR for BoPyV-1 on aliquots of the supernatant obtained at 5-, 4- and 5-days post-inoculation was positive with Ct values of 9.6, 13.2, and 14.3, respectively, demonstrating BoPyV-1 replication. No cytopathic effect was observed at the same time points in any of the MDBK cell cultures used as negative controls (Figure 6B).

## 4. Discussion

Here, we describe for the first time gross and microscopic lesions in an aborted bovine fetus infected with BoPyV-1, providing evidence of its pathogenicity under nonexperimental conditions in its natural host.

Altogether, the pathological (histological, immunohistochemical, ultrastructural) findings in the kidneys of the fetus are consistent with an active polyomavirus infection with lytic replication of the virus, production of abundant intranuclear inclusion bodies composed of dense arrays of assembled virions, and LT antigen expression in the nuclei of renal tubular epithelial cells. In addition, severe degeneration (i.e., swelling, tumefaction, attenuation) and necrosis of renal tubular epithelial cells are indicative of a cytopathic viral effect, along with a prominent inflammatory response to the infection, and interstitial fibrosis suggestive of chronic active renal damage at the time of abortion. While BoPyV-1 was isolated and detected by molecular methods, as well as intralesionally by immunohistochemistry and transmission electron microscopy, other DNA viruses able to produce intranuclear inclusion bodies in epithelial cells, such as herpesviruses and adenoviruses, were ruled out in this case. Although BoPyV-1 had been previously detected in aborted bovine fetuses [44], the study did not include a pathologic examination of the fetuses, and the authors considered BoPyV-1 an unlikely cause of abortion. Conversely, we provide evidence to consider that BoPyV-1 is fetopathogenic in its natural host. Elucidating whether the clinical manifestation of abortion resulted from BoPyV-1-induced pathology in the fetus needs further investigation, although based on the evidence provided herein, BoPyV-1 should be considered a probable cause of abortion in cattle.

Although the pathogenic mechanisms by which most polyomaviruses induce tissue damage are not yet fully understood, five patterns of polyomavirus-induced pathology have been proposed [45]. These include: (1)- cytopathic polyomavirus pathology pattern, (2)- cytopathic-inflammatory pathology pattern, (3)- immune-reconstitution inflammatory syndrome, (4)- autoimmune polyomavirus pathology pattern, and (5)- oncogenic polyomavirus pathology pattern. The cytopathic polyomavirus pathology pattern is characterized by uncontrolled viral replication in the infected cells without significant inflammation; PML caused by JCPyV replication in the oligodendrocytes is the prototype disease for this pattern. The cytopathic-inflammatory pathology pattern is characterized by high-level virus replication with cytopathic lysis of infected cells, along with necrosis and a significant inflammatory response. PyVAN in kidney allografts is the prototype disease for this pattern [45]. The lesions in the kidneys of the aborted fetus described herein best fit with this latter pattern of polyomavirus-induced pathology.

Histologically, PyVAN in humans is characterized by varying degrees of multifocal random interstitial nephritis, tubulitis, cytopathic changes and basophilic intranuclear inclusions in renal epithelial cells, interstitial fibrosis, and tubulo-interstitial atrophy, depending on the stage of disease progression [46]. Interestingly, these same lesions were observed in the kidneys of the aborted fetus infected with BoPyV-1. The resemblance in the type of renal lesions suggests that BoPyV-1 in cattle could potentially be a natural model of PyVAN, although this possibility needs to be further explored. Besides nephropathy and renal impairment, BKPyV has been occasionally associated with ureteral stenosis and hemorrhagic cystitis in humans [47]. Unfortunately, the ureters and urinary bladder of the BoPyV-1-infected bovine fetus were not subjected to pathologic examination.

Immunohistochemistry using antibodies against SV-40 LT antigen, as in this report, has been shown to cross-react with other polyomaviruses, such as BKPyV and JCPyV, in cases of PyVAN in humans [48]. In concert with this, the antibody we used cross-reacted with BoPyV-1, indicating that this test can be used for viral identification in formalin-fixed paraffin-embedded bovine tissues. This is not unexpected, considering that LT antigen has domains that are conserved among the polyomaviruses [1]. However, the LT antigen amino acid sequence identity between SV-40 and BoPyV-1/Faber/2021/Uy is only 37.4% (data not shown), although the similarity within the recognition sites of the antibodies (epitopes) may be higher.

The lesions observed in the hepatic parenchyma of the aborted fetus, including hepatocellular damage (i.e., scattered individual hepatocellular necrosis/apoptosis), inflammation, and fibrosis, are attributable to a chronic active infection. Although we were unable to clearly identify viral inclusions histologically or viral LT antigen by immunohistochemistry in the liver, real-time PCR results indicate that the viral genome was present in this tissue. Not finding viral inclusions nor antigen could be due to a lower limit of detection of these pathologic techniques compared to real-time PCR, or to a possible multifocal patchy distribution of the virus in the hepatic parenchyma. The real-time PCR Ct value found in the liver (14.8) would indicate a relatively high viral load in this tissue, though much lower than the one found in the kidney (Ct = 6.3). The molecular detection of the virus in the liver is consistent with the previous study from Belgium, in which kidney was not tested, but BoPyV-1 was detected in other fetal tissues (and fluids), including liver [44]. Our results indicate that the viral load is higher in the kidneys, where the virus can not only be detected by real-time PCR but also visualized by routine histopathology. Whether BoPyV-1 infects and replicates in other fetal tissues should be explored in future research.

Interestingly, histologic examination of the brain of the fetus revealed mild multifocal infrequent gliosis with rare perivascular and leptomeningeal lymphocytic infiltrates. These lesions were mild, infrequent, and likely incidental (sublethal). Given that the fetus was coinfected with *N. caninum*, as determined by PCR amplification of DNA of this protozoan parasite from the fetal brain, these brain lesions could be attributed to *N. caninum* infection. While *N. caninum* is a common abortifacient of cattle, abortions caused by this protozoan usually have typical and severe lesions that include multifocal encephalic necrosis along with gliosis and/or inflammation of the cerebral parenchyma, and extensive non-suppurative myocarditis and/or skeletal myositis [31], which were not present in this fetus. Hence, we believe that, while the mild brain lesions in the fetus could have been caused by this protozoan, other lesions typically found in fetuses aborted because of *N. caninum* infection were lacking. Multifocal mild cerebral gliosis and perivascular or leptomeningeal inflammation in the brain is occasionally present as an incidental (sublethal) finding in asymptomatic bovine neonates that are born congenitally infected with *N. caninum* or non-aborted infected fetuses recovered at slaughter [49,50,51]. Interestingly, somewhat similar lesions including gliosis and perivascular lymphocytic encephalitis have been recently described in two adult dairy cows infected with BoPyV-2 [21]. Considering that BoPyV-1 was detected in the brain of the fetus by real-time PCR, although with a higher Ct (19.9) indicating a lower viral load than in the kidney and liver, the contribution of the virus in the development of these cerebral lesions should not be disregarded. However, whether BoPyV-1 contributed to these cerebral lesions, or whether *N. caninum* contributed to the clinical presentation of abortion, remains under speculation. Of note, *N. caninum* PCR was negative in the fetal kidney and liver, suggesting that this parasite did not contribute to the lesions observed in these tissues.

Generally, occurrence and progression of diseases caused by polyomaviruses in mammals seem to be largely dependent on host immunosuppression [1,2]. Pregnant females can experience modulation in their immune system making them more prone to pathogenic infections [52]. While the real-time PCR performed on serum of the cow indicates that she was viremic at the time of the abortion, whether she or the aborted fetus in this report were immunosuppressed is unknown. There was no evidence of lymphoid depletion on histologic examination of lymphoid tissues including thymus, spleen, and lymph nodes in the aborted fetus. Bovine viral diarrhea virus (*Pestivirus*), a common immunosuppressive virus of cattle, was ruled out by reverse transcriptase PCR in the fetal tissues. Additionally, *Leptospira* spp. infection, which can also cause bovine abortion with fetal renal and hepatic lesions, was ruled out by real-time PCR in the kidney and liver.

Little information is available on the epidemiology and geographic distribution of BoPyVs. An early serologic study found that 62% of 273 cattle had antibodies against the virus [18], suggesting a relatively high seroprevalence. Based on the authors’ affiliations, we speculate that the tested cattle were from the United Kingdom. Similarly, a research group from The Netherlands found antibodies to the virus in sera of 25 out of 57 cattle tested (43.9%), as well as in 6/26 (23.1%) samples of bovine colostrum [9]. A study from New Zealand found that infection is more common in bovine fetuses and calves than in adult cattle when batches of bovine serum products are analyzed [15]. Studies based on molecular detection in bovine serum, beef muscle, or ground beef in the USA, Mexico, Germany, and New Zealand found DNA of BoPyV-1, -2, and/or -3 in 2–70% of the tested samples [15,16,20,53,54]. Molecular detection of BoPyV-1 and/or -2 has also been documented in cattle in Spain [55], Belgium [44], and Switzerland [21]. BoPyV has also been identified as an environmental contaminant in Spain [56,57], Greece, Hungary, Sweden, and Brazil [56]. The name of the initial polyomavirus isolated from cattle designated WRSV suggests that this virus may have been isolated in Wokalup Research Station in Australia [17]. Our report broadens the current knowledge on the geographic distribution of BoPyV-1 to Uruguay. Although the available information is somewhat limited, BoPyVs seem to have a broad geographic distribution.

In this work, we sequenced the whole genome of the involved BoPyV-1, which, to the best of our knowledge, represents the first available genome from the Southern Hemisphere. However, we note that one of the BoPyV-1 complete genomes available in GenBank (D13942) lacks data on its geographic origin, information that is also missing in the original publications describing this isolate and sequence by researchers from The Netherlands [9,19]. The genome of all known polyomaviruses encodes at least two regulatory proteins, namely LT and ST antigens, and two structural proteins, the major capsid protein VP1 and the minor capsid protein VP2 [1]. A third capsid protein (VP3) is encoded by most polyomaviruses. Similar to other members of the *Polyomaviridae* family, such as BKPyV and JCPyV, BoPyV-1 also encodes the regulatory protein agnoprotein which is required for efficient viral proliferation. Accordingly, BoPyV-1/Faber/2021/Uy genome encodes all the previously mentioned proteins.

Phylogenetic analyses revealed a close relationship of BoPyV-1/Faber/2021/Uy to the other BoPyV-1, particularly with the first complete genome released in GenBank (D13942) [19]. This close relationship was also observed in the analyses of sequence identity, in which BoPyV-1/Faber/2021/Uy showed an identity higher than 99% to D13942 when comparing the complete nucleotide sequence and also at both the nucleotide and amino acid level. It is worth mentioning a particular event that was observed in the ST antigen gene, where isolate H8 (KX455485) does not present a proper coding sequence of this gene, as observed in 55% of the reads composing the BoPyV-1/Faber/2021/Uy genome. This event is caused by a T/A substitution that generates a stop codon, which may indicate that the ST antigen is not essential for virus replication and disease. It should not be overlooked that this substitution in 55% of the reads could represent a sequencing artifact.

Despite not being an objective of this work, based on the obtained results and the latest release of the ICTV classification system by the *Polyomaviridae* Study Group, we propose to assign BoPyV-2 and BoPyV-3 in the genera and species *Alphapolyomavirus secubovis* and *Deltapolyomavirus tertibovis*, respectively.

Finally, we isolated the BoPyV-1 infecting this fetus from a frozen sample of kidney. Based on the results of the histologic and ultrastructural examinations, as well as immunohistochemistry, which indicated active viral replication in renal tubular epithelial cells, we elected to attempt virus isolation on MDBK cells, as they are epithelial cells of bovine kidney origin. Surprisingly, under the conditions we described, cytopathic effect was evident 4–5 days after inoculating the cells with either the fetal kidney that had been kept frozen for nearly 1 year, or the supernatants of the first and second passages. Concurrently, high BoPyV-1 loads were identified in the culture supernatants, as determined by the low real-time PCR Ct values. Altogether the results indicate that BoPyV-1 remains viable for long periods under freezing, as expected for a non-enveloped single stranded DNA virus, and that this strain may have a high replication capacity, considering that BoPyV-1 has been regarded as a relatively slow-growing virus, requiring at least 3 to 5 weeks before in vitro virus replication is detected by qPCR [14,58].

## 5. Conclusions

We conclude that BoPyV-1 is pathogenic to the bovine fetus and, thus, a probable cause of abortion in viremic cattle. Pathogenicity seems to involve cytopathic viral effects with tissue damage including necrosis, inflammation, and fibrosis, with lesions resembling PyVAN in humans. Factors of the virus and host that would determine pathology and disease development, as well as the epidemiology and transmission routes and zoonotic potential, need further investigation. In vitro and in vivo studies, including experimental infections in laboratory animals and livestock using the BoPyV-1 strain isolated from this fetus would help to better understand the biology and clinical relevance of this virus.

## Figures and Tables

**Figure 1 viruses-14-02042-f001:**
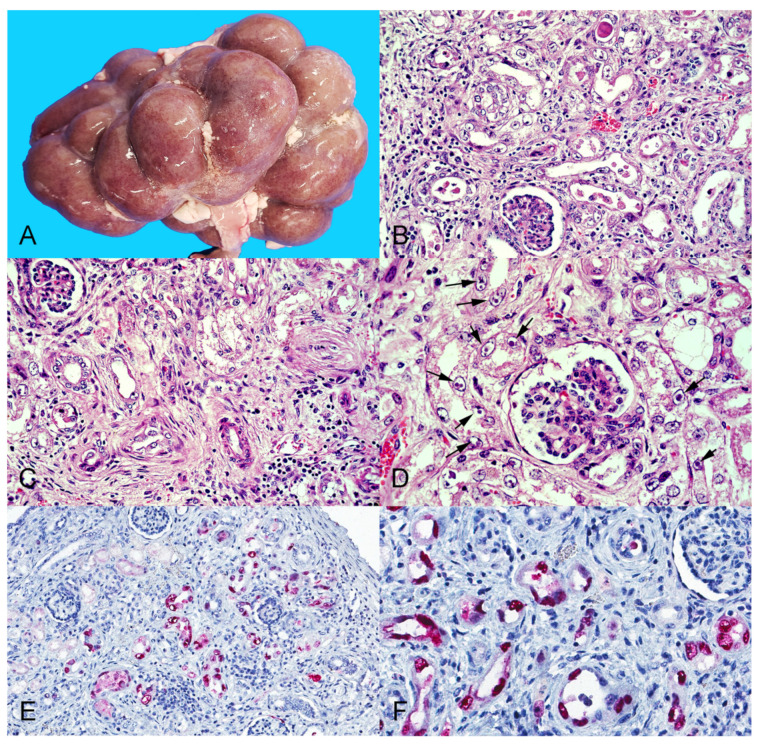
Pathological findings in the kidney of the aborted fetus. (**A**) Grossly, there are numerous discrete pinpoint red foci disseminated throughout the renal cortex that are visible through the capsular surface. (**B**) Cortical tubules are variably ectatic, contain necrotic cellular debris and are lined by either attenuated epithelium, or epithelial cells with markedly swollen vesicular nuclei, occasionally containing one or several magenta intranuclear inclusion bodies. The cortical interstitium is infiltrated by inflammatory cells, predominantly lymphocytes and histiocytes. Hematoxylin and eosin (HE) stain, original magnification 400×. (**C**) Similar histologic lesions as described in B but there is also marked interstitial fibrosis. HE stain, original magnification 400×. (**D**) Numerous cortical tubular epithelial cells have markedly swollen nuclei containing one or several magenta intranuclear inclusion bodies surrounded by a clear halo (arrows), indicating margination of the chromatin. HE stain, original magnification 630×. (**E**,**F**) Immunoreactivity to polyomavirus LT antigen is demonstrated by red chromogen precipitate largely in the nucleus of cortical tubular epithelial cells. Immunohistochemistry (immunoalkaline phosphatase technique) with a monoclonal primary antibody against SV-40 polyomavirus LT antigen, hematoxylin counterstain, original magnifications 200× (**E**) and 400× (**F**).

**Figure 2 viruses-14-02042-f002:**
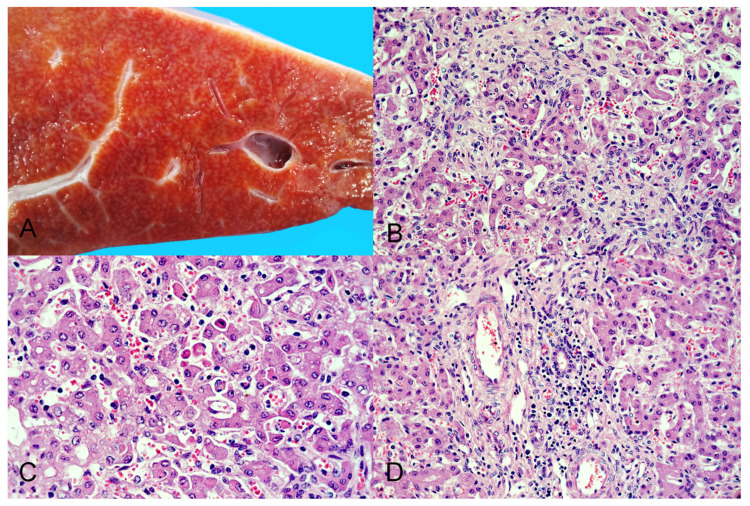
Pathological findings in the liver of the aborted fetus. (**A**) Cut section of the hepatic parenchyma showing an enhanced reticular pattern characterized by numerous pinpoint pale grayish foci and interconnected linear streaks alternating with reddish-orange areas of hepatic parenchyma (“nutmeg liver”). (**B**) The hepatic histoarchitecture is distorted and hepatic cords are disorganized and separated by areas of fibroplasia characterized by spindle cells (fibroblasts) embedded in a pale eosinophilic fibrillar (collagenous) extracellular matrix. HE stain, original magnification 400×. (**C**) Some hepatocytes are individualized and detached from the hepatic cords, and are shrunken, with angular cell borders, hypereosinophilic cytoplasm and pyknosis or karyorrhexis (hepatocellular necrosis/apoptosis); others have one or few clear intracytoplasmic vacuoles consistent with lipidosis. There are increased numbers of lymphocytes in the sinuses (hepatitis). HE stain, original magnification 630×. (**D**) A portal tract (center) is infiltrated by moderate numbers of lymphocytes and macrophages (portal hepatitis). HE stain, original magnification 400×.

**Figure 3 viruses-14-02042-f003:**
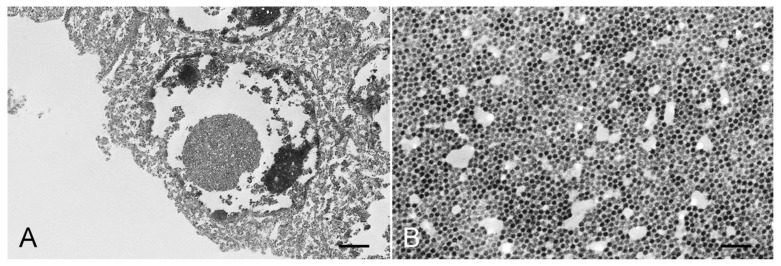
Transmission electron microscopy in the kidney of the aborted fetus. (**A**) The nucleus of a tubular epithelial cell contains a dense array of viral particles forming a round inclusion (7190×, bar = 2 µm). (**B**) Higher magnification of A showing the array of viral particles (70,000×, bar = 200 nm).

**Figure 4 viruses-14-02042-f004:**
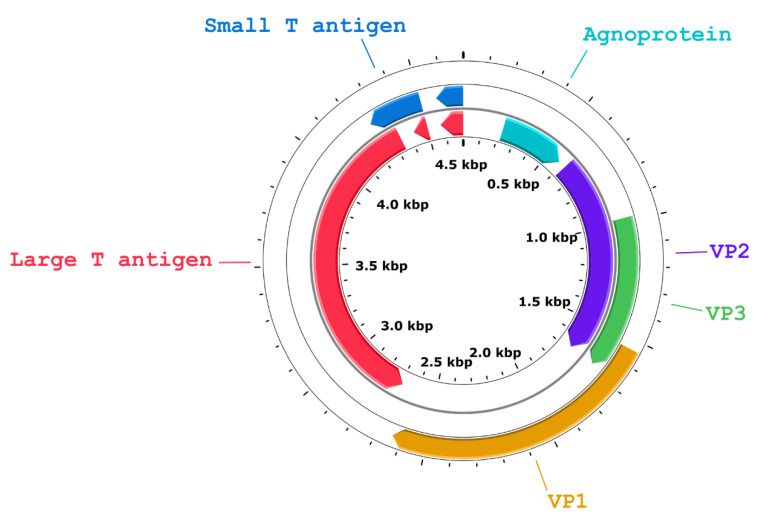
Genome organization of BoPyV-1/Faber/2021/Uy (GenBank accession number OM938033). The different proteins are represented with arrows according to their location in the genome; arrow direction indicates the strand. The agnoprotein (position 220–250), VP1 (position 1540–2637), VP2 (position 618–1679), and VP3 (position 981–1679) are located on one strand and the LT antigen (joined positions 2690–4345, 4423–4502, and 4574–4697), and the ST antigen (joined positions 4252–4502 and 4574–4697) are located on the complementary strand. The complete genome length of BoPyV-1/Faber/2021/Uy is 4697 base pairs.

**Figure 5 viruses-14-02042-f005:**
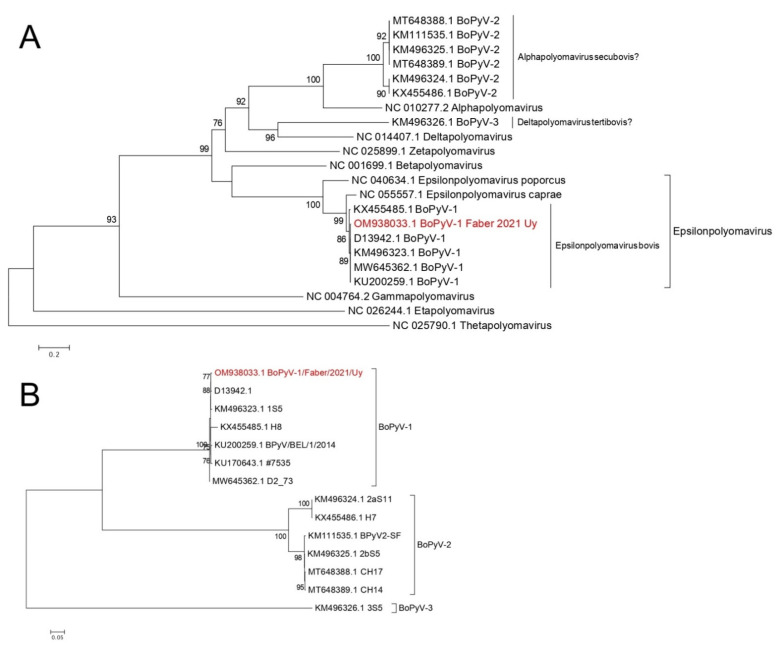
Phylogenetic analyses. (**A**) Phylogenetic tree for LT antigen amino acid sequences. Representative sequences of all ICTV recognized genera, all polyomavirus sequences of bovine origin from GenBank, and the sequence obtained in this work (in red font) were used. Sequences were aligned with ClustalW. The best substitution model (rtREV+F+I+G4) and the maximum likelihood phylogenetic tree were obtained with IQ-TREE web server. Branch support analysis was SH-aLRT branch test implemented in the IQ-TREE web server (1000 replicates). (**B**) Phylogenetic tree using complete genome nucleotide sequences. All polyomavirus sequences of bovine origin from GenBank and the sequence obtained in this work (in red font) were used. Sequences were aligned with ClustalW, and the best substitution model (K3Pu+F+G4) and the maximum likelihood phylogenetic tree were jointly obtained with IQ-TREE web server. Branch support analysis was SH-aLRT branch test implemented in the IQ-TREE web server (1000 replicates).

**Figure 6 viruses-14-02042-f006:**
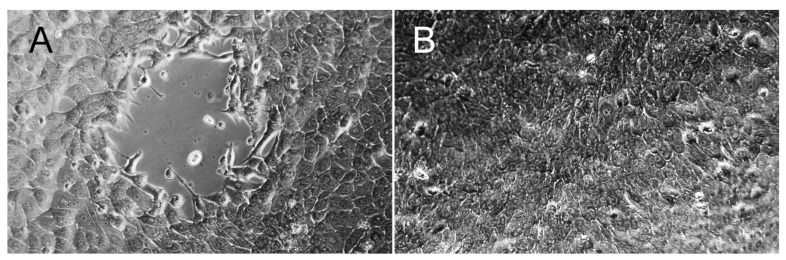
BoPyV-1 isolation in MDBK cells. (**A**) Cytopathic effect after 96 h of incubation is characterized by cytoplasmic vacuolation and lysis with loss of the monolayer (center) in infected MDBK cells. (**B**) In the negative control, uninfected MDBK cells form a continuous monolayer. Original magnifications 200×.

**Table 1 viruses-14-02042-t001:** Comparison between BoPyV-1/Faber/2021/Uy and members of the *Epsilonpolyomavirus bovis* (BoPyV-1), BoPyV-2, and BoPyV-3 at the nucleotide and amino acid levels.

		*Epsilonpolyomavirus bovis* (BoPyV-1)		BoPyV-2		BoPyV-3	
		Nucleotide	Amino Acid	Nucleotide	Amino Acid	Nucleotide	Amino Acid
**BoPyV-1/Faber/2021/Uy**	Complete genome	93.6–99.3	-	40.0–41.0	-	37.6	-
	Large T antigen	93.8–99.3	94.1–99.6	36.1–37.5	30.3–31.6	48.3	36.9
	Small T antigen	99.2–99.7	98.3–99.1	26.5–28.9	15.7–17.1	26.7	18.2
	VP1	97.1–99.3	98.6–99.7	53.7–55.0	50.5–52.2	39.7	27.5
	VP2	97.0–99.3	96.8–99.7	31.0–32.1	23.7–25.1	31.8	16.4
	VP3	96.1–99.2	96.1–100	36.1–37.7	20.6–22.1	30.1	13.0

## Data Availability

Not applicable.

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
