# Peer review of "Bovine Polyomavirus-1 (Epsilonpolyomavirus bovis): An Emerging Fetal Pathogen of Cattle That Causes Renal Lesions Resembling Polyomavirus-Associated Nephropathy of Humans"

_viruses, 2022, doi:10.3390/v14092042_

Round 1
Reviewer 1 Report
The work reports on the identification of Bovine polyomavirus-1 (BoPyV-1) in the an aborted bovine fetus, providing evidence of pathogenicity and probable abortigenic potential for this polyomavirus. The main findings indicate that: the virus can cause severe kidney lesions in cattle attributable to active viral replication in renal tubular epithelial cells.
Authors also identified the presence of the BoPyV-1 oncoprotein Large T antigen renal tubular epithelial cells as well as the presence of productive BoPyV-1 replication and virion assembly in the nuclei of renal tubular epithelial cells
The ms is in general well written, and the reported data are interesting. The present findings will improve our knowledge on polyomaviruses, in this case bovine polyomaviruses, and their infective capacity, which is an interesting topic.
In other words, the work is well written and well covers the subject. The work is suitable for publication in viruses MDPI.
However, several improved should be made. Please find enclosed several observations for improving the manuscript.
Major
1. It would be interesting investigating viral footprints in the dam in the hypothesis of a vertical transmission of the BoPyV-1. Have the authors any information on this issue? And what about placenta tissues? Was the BoPyV-1 DNA been investigated in this such of tissue?
2. BKPyV infection often is associated with renal impairment, including ureteral stenosis, hemorrhagic cystitis, and nephropathy (doi: 10.3390/jcm8091477 and doi: 10.2215/CJN.02770707) . Authors should discuss these notions in relation to their findings obtained with BoPyV-1 and renal tissues/cells. A similar genomic/protein composition and viral activity which lead to a similar kidney tropism cannot be excluded for these two polyomaviruses.
Minor
1. Lines 51-52 I would include humans
2. Line 72 a recent paper suggest that Merkel cell polyomavirus can possibly be related to abortive events in humans (PMID: 34970247). This is an important observation that may support the findings of the present manuscript. This notion and supporting reference should therefore be included
3. Lines 95-71 and line 565, besides BK and JC, Merkel cell polyomavirus oncogenic activity have bene reported to increase in conditions of iatrogenic immune suppression DOI: 10.1158/1078-0432.CCR-16-2899 and DOI: 10.1111/apm.12122
4. Lines 194 and 199 BoPyV-1 VP1 gene?
5. Lines 530-352 As the authors have isolated the complete genomic sequence of BoPyV-1, a generation of a standard curve with different concentrations of BoPyV-1 DNA fore determining the viral DNA load is suggested. This might be more informative than reporting the qPCR ct alone
6. Line 566 Pregnant females can experience modulations in their immune system making them more prone to pathogenic infections (doi: 10.1093/humupd/dmv041). So instead of immunosuppression we can argue in favor for a general and physiological immune adaptation/modulation process occurring in pregnancy to hamper fetus rejection
Reviewer 2 Report
Gianittti et al have shown pathological evidence that Bovine polyomavirus-! (BoPyV1) is a possible causative agent for fetus abortion in cattle in this work. The authors found that kidney lesions in an aborted fetus were positive for BoPyV1 Large T antigen (LT) and furthermore showing typical polyomavirus presence based on microscopic studies. The investigators sequenced the BoPyV1 genome of the isolated virus and tested the virus in MDBK cells.
The results are based on a one case. It would be better to analyze another, or more aborted fetuses include in this study.
Figure 4 should be re-drawn more professionally, and location of each viral gene as well as the regulatory region should be indicated on the genomic map based on its location.
Have the investigators performed immunocytochemistry of the MDBK cells to demonstrate the presence of Large T in the cell nucleus?
